# Local-Peak Scale-Invariant Feature Transform for Fast and Random Image Stitching

**DOI:** 10.3390/s24175759

**Published:** 2024-09-04

**Authors:** Hao Li, Lipo Wang, Tianyun Zhao, Wei Zhao

**Affiliations:** 1State Key Laboratory of Photon-Technology in Western China Energy, International Collaborative Center on Photoelectric Technology and Nano Functional Materials, Institute of Photonics & Photon Technology, Northwest University, Xi’an 710127, China; 202232601@stumail.nwu.edu.cn; 2UM-SJTU Joint Institute, Shanghai Jiao Tong University, Shanghai 200030, China; lipo.wang@sjtu.edu.cn; 3School of Automation, Northwestern Polytechnical University, Xi’an 710072, China; zhaoty@nwpu.edu.cn

**Keywords:** image stitching, LP-SIFT, image mosaic

## Abstract

Image stitching aims to construct a wide field of view with high spatial resolution, which cannot be achieved in a single exposure. Typically, conventional image stitching techniques, other than deep learning, require complex computation and are thus computationally expensive, especially for stitching large raw images. In this study, inspired by the multiscale feature of fluid turbulence, we developed a fast feature point detection algorithm named local-peak scale-invariant feature transform (LP-SIFT), based on the multiscale local peaks and scale-invariant feature transform method. By combining LP-SIFT and RANSAC in image stitching, the stitching speed can be improved by orders compared with the original SIFT method. Benefiting from the adjustable size of the interrogation window, the LP-SIFT algorithm demonstrates comparable or even less stitching time than the other commonly used algorithms, while achieving comparable or even better stitching results. Nine large images (over 2600 × 1600 pixels), arranged randomly without prior knowledge, can be stitched within 158.94 s. The algorithm is highly practical for applications requiring a wide field of view in diverse application scenes, e.g., terrain mapping, biological analysis, and even criminal investigation.

## 1. Introduction

Image stitching is employed to reconstruct complete image information from the image fragments [1]. Due to the limited capability to capture a large area with high spatial resolution, image stitching finds widespread uses in engineering applications such as machine vision [2,3,4,5], augmented reality [6,7,8,9], navigation [10,11], and panoramic shooting [12,13,14]. Furthermore, it plays an indispensable role in supporting scientific research, such as the construction of large-scale bioimages [15,16,17,18,19], micro-/nanostructures [20,21,22], aerial photography [23,24,25], and space remote sensing [26].

Various image stitching techniques can be traced back to the late 1970s or early 1980s [27,28,29], aiming to merge multiple images to achieve a wider field of view. In these early stages, multiple images of the same scene were simply superimposed [1]. At present, advanced image stitching techniques can be primarily categorized into two categories: region-based and feature-based [28]. In comparison to the computationally intensive region-based stitching techniques [13], feature-based techniques impose fewer requirements on region overlapping [30], rendering them increasingly appealing in recent years. Harris et al. [31] initially proposed a feature-based stitching technique for detecting features in images based on corner points. Subsequently, Lowe et al. [32] introduced the scale-invariant feature transform (SIFT) approach for feature detection, utilizing the Gaussian pyramid and Gaussian difference pyramid. Although SIFT has a series of features, such as robustness, scale invariance, and rotation invariance, it necessitates substantial computational resources, leading to a slower feature point detection and accordingly, lower stitching speed.

To address the limitations of SIFT, Rosten et al. [33] proposed the Features from Accelerated Segment Test (FAST), a machine learning algorithm for high-speed corner detection. However, FAST sacrifices scale invariance and rotation invariance features. To mitigate these drawbacks, Bay et al. [34] proposed the Speeded Up Robust Features (SURF) algorithm in 2008, which introduced a new scale and rotation invariant detector and descriptor. While SURF offers scale invariance, and rotation invariance, it is less robust to viewpoint changes. In 2011, Rublee et al. [35] presented an efficient alternative to SIFT and SURF, named Oriented FAST and Rotated BRIEF (ORB). However, it is susceptible to variations in illumination. In the same year, Stefan et al. [36] proposed the Binary Robust Invariant Scalable Keypoints (BRISK) method. Similar to SIFT, BRISK predominantly utilizes FAST9-16 to identify feature points and generates feature descriptor vectors in binary form, which exhibit sensitivity to variations in illumination and rotation. Later, to overcome boundary blurring and detail loss in SIFT and SURF, Alcantarilla et al. [37] proposed the KAZE algorithm in 2012, which can preserve more details by constructing a nonlinear scale space to detect feature points. However, similar to SIFT, the KAZE algorithm also requires a large calculation time. In 2013, Pablo et al. [38] proposed the Accelerated KAZE (AKAZE) algorithm to improve the calculation efficiency by changing the description of feature vectors.

Although in recent years the research focuses have shifted more toward the deep learning neural network [39,40,41], it is still effective to optimize and update the conventional algorithms for image stitching applications. For instance, the parity algorithm by Wu et al. [42], as an improvement of SIFT, can reduce the matching error with better accuracy. The segmentation algorithm by Gan et al. [43] helps to improve the stitching speed and the SIFT accuracy. An image registration method [44] combines Shi-Tomasi with SIFT to improve the matching accuracy and reduce the computational cost.

In feature-based image stitching techniques, the precise definition and detection of feature points are pivotal. The algorithm’s efficiency can be significantly enhanced if there is a reasonable replacement for the Gaussian pyramid and Gaussian difference pyramid operations in SIFT. In the dissipation element (DE) analyses of turbulent flows [45,46], the dynamic nature of the field can be characterized by the statistical properties of the extremal points. Moreover, the concept of extremal points can be extended to the multiscale level [47]. From a geometric standpoint, any image can be viewed as a structure with multiscale intensity, akin to the identified DEs. Considering the multiscale nature of the DE analysis in turbulent flow field and that in image analysis, it is possible to replace the Gaussian pyramid in the SIFT algorithm by the extremal points in DE analysis, to improve the efficiency of feature point detection in the SIFT algorithm.

In this study, we aim to construct a fast feature point detection algorithm, termed the local-peak scale-invariant feature transform (LP-SIFT), which integrates SIFT with the concept of local extremal points or image peaks at the multiscale level. Specifically, the featured points can be substituted with the multiscale local peak points without reliance on the Gaussian pyramid and differential pyramid in SIFT. Consequently, the efficiency of feature point detection can be significantly enhanced. Additionally, by combing LP-SIFT and RANSAC in image stitching, the stitching speed can be notably improved, particularly in the processing of large images, when compared with the original SIFT method. The framework of the algorithm is briefly introduced in Figure 1. Finally, we demonstrate that random images or segments of a large image can be successfully stitched within an acceptable time without the requirement of prior knowledge.

## 2. Materials and Methods

The principle of the LP-SIFT method is schematically diagramed in Figure 2. Generally, there are five sub-steps, namely, image preprocessing, feature point detection, feature point description, feature point matching, and image stitching. We will initially provide a detailed introduction to each step. Subsequently, a strategy of applying LP-SIFT to restore a large image from random image fragments will be elucidated.

### 2.1. Image Preprocessing

In the following exemplified case, the two images (Figure 2a) designated for stitching are referred to as the reference image (stored as the reference matrix $M_{1}$) and registered image (stored as the registration matrix $M_{2}$). Since the feature points adopted in LP-SIFT are the local peak points (both maximum and minimum), to avoid the difficulty of locating them in the constant image intensity region, e.g., due to saturation, it is numerically meaningful to first impose a small linear background on both images. Therefore, the new image matrixes $M_{n,k}$ becomes
(1)$$M_{n,k}\left(i,j\right)=M_{k}\left(i,j\right)+\left[\left(i-1\right)*nc_{k}+j\right]*\alpha$$
where α≪1 is the linear noise coefficient. Now the size of $M_{n,k}$ is $nr_{k}\times nc_{k}$, $nr_{k}$ is the row of the $M_{n,k}$, and $nc_{k}$ is the column of the $M_{n,k}$, where k=1 represents parameters of the reference image and k=2 represents parameters of the registered image.

### 2.2. Feature Point Detection

In step 2, we utilize the local peak points of multiscale images as the feature points (Figure 2b). Both the reference image and the registered image are partitioned into squared interrogation windows of size L, with varying scales, e.g., L=32 to 128. The maximum and minimum points in each interrogation window are collected as feature points, which can be formulated as
(2)$$\left\{\begin{array}{c} M_{n,k,\max}\left(p,L\right)=\max\left[M_{np,k}\left(i,j\right)\right],\;i,j\in [0,L]\\ M_{n,k,\min}\left(p,L\right)=\min\left[M_{np,k}\left(i,j\right)\right],\;i,j\in [0,L] \end{array}\right.$$
where $M_{np,k}\left(i,j\right)$ is the *p*th interrogation window of L×L pixels.

### 2.3. Feature Point Description

Once all the feature points are gathered, the SIFT feature description vector is employed to characterize the acquired feature points. First, as shown in Figure 2d, around each feature point (represented by a red spot), a square region with the size w is extracted, with
(3)$$\begin{array}{c} w=\beta Ld \end{array},$$
where $\beta ={\beta_{0}\left(L/L_{max}\right)}^{-1/2}$ is an adjustable factor to control the size of the square region and $\beta_{0}$ is the initial β when the interrogation window size L equals its maximum value $L_{max}$ (e.g., 128). To ensure w is not over L, it requires
(4)$$\beta_{0}\leq \frac{1}{d}{\left(\frac{L}{L_{max}}\right)}^{-\frac{1}{2}}.$$

We further divide the square region into d×d subregions, e.g., d=4, as commonly used [48]. In each subregion $M_{n,k,s}$, there are $w_{s}\times w_{s}$ pixels, with $\begin{array}{c} w_{s}=\beta L \end{array}$ as the side length of the subregion. At each pixel $\left(i,j\right)$ in this subregion, the magnitude $f_{k}\left(i,j\right)$ and orientation $\theta_{k}\left(i,j\right)$ of the image gradient can be calculated as
(5)$$\left\{\begin{array}{c} f_{k}\left(i,j\right)=\sqrt{a^{2}+b^{2}}\\ \theta_{k}\left(i,j\right)=b/a\\ a=M_{n,k,s}\left(i+1,j\right)-M_{n,k,s}\left(i-1,j\right)\\ b=M_{n,k,s}\left(i,j+1\right)-M_{n,k,s}\left(i,j-1\right) \end{array}\right.$$

Then, the distribution of the image gradient in 8 directions [48] can be determined based on $\theta_{k}\left(i,j\right)$ and $f_{k}\left(i,j\right)$. After the calculation in eight directions for each subregion, a 128-element array can be obtained, i.e., d×d×8 when d=4. Such a 128-element array is then used as the feature description vector ($\vec{d_{f}}$) of the feature points [32]. Due to the statistical determination of the array over a wide area, the matching robustness is maintained.

### 2.4. Feature Point Matching

The position information (i.e., the pixel coordinates of the feature points) and feature description vector of each feature point in the reference images are denoted as $\vec{p_{v1}}$ and $\vec{d_{f1}}$, respectively. The corresponding ones in the registered image are denoted as $\vec{p_{v2}}$ and $\vec{d_{f2}}$. Then, the difference (∆) between the feature description vectors is
(6)$$∆=\sqrt{{\left|\vec{d_{f1}}-\vec{d_{f2}}\right|}^{2}}.$$

The smaller the difference (∆) between the two feature description vectors, the more similar they are. The preset threshold of ∆ is denoted as ${∆}_{s}$. Clearly, ${∆}_{s}$ signifies the requirement for similarity between the feature description vectors, thereby determining the number of matching points to be retained. Only when $∆<{∆}_{s}$, the $\vec{p_{v1}}$, $\vec{p_{v2}}$, $\vec{d_{f1}}$, and $\vec{d_{f2}}$ of the matched pairs are stored for image stitching.

### 2.5. Image Stitching

Based on the matched pairs obtained previously, the imaging stitching process utilizes the Random Sample Consensus (RANSAC) algorithm [49]. RANSAC operates under the fundamental assumption that the sample comprises both accurate data (inliers, data that conform to the model) and abnormal data (outliers, data that deviate from the expected range and do not align with the model), which may, for instance, result from noise [50] and improper measurements, assumptions, or calculations. Furthermore, RANSAC also assumes that for a given accurate dataset, the model parameters can be consistently computed.

As shown in Figure 2f, a homography matrix (H) is employed to depict the perspective transformation of a plane in the real world along with its corresponding image. This matrix is utilized to facilitate the transformation of the image from one viewpoint to another through the perspective transformation process. Therefore, the relationship between the matched pairs can be obtained as follows
(7)$$\left[\begin{array}{c} x^{\prime }\\ y^{\prime }\\ 1 \end{array}\right]=H\left[\begin{array}{c} x\\ y\\ 1 \end{array}\right]$$
where $\left(x,y\right)$ represents the pixel coordinates of the matched points in the reference image. $\left(x^{\prime },y^{\prime }\right)$ represents the pixel coordinates of the matching points in the registered image. H can be expressed as
(8)$$H=\left[\begin{array}{ccc} \cos\theta & -\sin\theta & t_{x}\\ \sin\theta & \cos\theta & t_{y}\\ 0 & 0 & 1 \end{array}\right]$$
where θ represents the angular difference between the reference image and the registered image. $t_{x}$ and $t_{y}$ represent the translational difference between the reference image and the registered image. All the quantities in Equation (8) are determined through the matched image pairs.

## 3. Results of Stitching on Two Images

In this section, we present a performance evaluation of the LP-SIFT algorithm, along with comparative results from other algorithms, such as SIFT, ORB, BRISK, and SURF. Table 1 presents different hardware and software resources used in this study. The code of the LP-SIFT algorithm was written in MATLAB 2021a with parallel computing. The code of SIFT [51] was downloaded from the internet. The code packages of ORB, SURF, and BRISK are from the Computer Vision Toolbox of MATLAB.

To ensure consistent comparison across different scenarios, the SIFT, ORB, BRISK, SURF, and LP-SIFT algorithms were utilized to compute feature points and feature description vectors in the two images. Subsequently, the RANSAC algorithm was employed to stitch the images together.

### 3.1. Datasets

To evaluate the performance of LP-SIFT algorithm, a range of datasets that encompass various scenarios, pixel sizes, and different levels of distortions were studied. These datasets include a rich assortment of illumination conditions and structural features in both natural and artificial environments. In this study, two datasets were used: Dataset-A and Dataset-B. Dataset-A contains three image pairs commonly adopted in relevant studies, namely, (1) mountain [52], (2) street view [53], and (3) terrain [54]. Dataset-B is captured by a camera of a mobile phone (PGKM10, OnePlus) with a resolution of 6 Mega-Pixels. It contains three image pairs: (1) building, (2) campus view (translation), and (3) campus view (rotation). The image pairs of the datasets are shown in Figure 3 as examples.

### 3.2. Evaluation Metrics

Since the datasets selected in this study were all captured from actual scenes without a reference image, it is difficult to use structural similarity (SSIM) [55] and the peak signal-to-noise ratio (PSNR) [56] to evaluate the image stitching results. To compare the stitching results of different methods, indicators such as average gradient (AG) [57] and spatial frequency (SF) [58] are employed.

AG reflects the detail contrast and texture transformation in the image and can be used to evaluate the quality of the fused image. In image stitching, the larger the AG, the better the stitching quality. AG can be defined as follows [57]:(9)$$\nabla G=\frac{1}{c*r}\sum_{i=1}^{c}\sum_{j=1}^{r}\sqrt{\frac{{\left(M\left(i+1,j\right)-M\left(i-1,j\right)\right)}^{2}+{\left(M\left(i,j+1\right)-M\left(i,j\right)\right)}^{2}}{2}}$$
where c represents the number of rows of the image and r represents the number of columns of the image.

SF is another evaluation metric reflecting the change rate of the image gray level. The larger the SF, the clearer the image, particularly for the image fusion after stitching. SF is calculated as follows [58]:(10)$$SF=\sqrt{\frac{1}{c*r}\sum_{i=1}^{c}\sum_{j=1}^{r}\left|M\left(i,j\right)-M\left(i,j-1\right)\right|+\frac{1}{c*r}\sum_{i=1}^{c}\sum_{j=1}^{r}\left|M\left(i,j\right)-M\left(i-1,j\right)\right|}$$

### 3.3. Images of Small Size

Small-sized images are commonly employed in various fields such as medical imaging [18], industrial inspection [59,60,61], and bridge inspection [62]. Hence, the performance of image stitching, which combines LP-SIFT and RANSAC, is initially assessed for these small-sized images. In our investigation, the images in the datasets of mountain and street view have small sizes. The sizes of the images are 602 × 400 pixels and 653 × 490 pixels, respectively. The stitching results are collectively shown in Figure 4, incorporating those obtained using SIFT, ORB, BRISK, SURF, and LP-SIFT. The corresponding parameters of the stitching process are summarized in Table 2. It is evident that all the five feature point detection algorithms, when combined with RANSAC can successfully stitch the two images. We computed the AG and SF of the stitching results. While the SIFT algorithm yielded the highest value for the mountain dataset and the ORB algorithm yielded the highest value for the street view dataset, the differences among the values of the five algorithms are very small. The stitching effect across the various algorithms is comparable. However, there are significant differences in computation times. It should be noted that the computation time encompasses feature point detection, feature description vector calculation, pair matching, and image stitching.

For the mountain dataset, the computation times are 101.21 s for SIFT, 0.71 s for ORB, 1.30 s for BRISK, 0.85 s for SURF, and 1.16 s for LP-SIFT. ORB takes the least time, while SIFT takes the most time. The comparison is clearer in Figure 5. However, for the street view dataset, the computation times are 226.62 s for SIFT, 2.27 s for ORB, 2.22 s for BRISK, 2.54 s for SURF, and 2.05 s for LP-SIFT. LP-SIFT takes the least time, while SIFT takes the most time. Furthermore, the stitching outcomes produced by the SIFT algorithm exhibit notable misalignment at the seams, whereas the other algorithms maintain alignment without noticeable discrepancies. In summary, SIFT is the most computationally intensive, with occasional misalignment in stitching results, while the stitching time of LP-SIFT, ORB, BRISK, and SURF are on the same level.

### 3.4. Images of Medium Size

For the purpose of high-quality visual presentation of photos [63], videos [64], and other scenes, images of 1080P (1080 × 1920 pixels) are commonly utilized. In our dataset, the image pairs in the datasets of terrain and building have a medium size. The former has a size of 1024 × 768 pixels and the latter are 1080 × 1920 pixels.

The stitching results are collectively shown in Figure 6, incorporating those obtained using SIFT, ORB, BRISK, SURF, and LP-SIFT. The corresponding parameters of the stitching process are summarized in Table 2. For the terrain dataset, the five feature point detection algorithms, when combined with RANSAC, can successfully stitch the two images. The SIFT algorithm takes up to 1674.87 s to accomplish the stitching; in contrast, ORB, BRISK, SURF, and LP-SIFT algorithms take 15.77 s, 3.20 s, 5.16 s and 4.47 s, respectively. Furthermore, the LP-SIFT algorithm provides the highest evaluation metrics in determining the parameters for stitching results.

For the building dataset, only four feature point detection algorithms, when combined with RANSAC, can successfully stitch the two images. Although the SURF algorithm yielded the highest evaluation metrics, the differences among the values of the four algorithms are within 2%, which is negligible; the ORB algorithm takes 327.25 s, the BRISK algorithm takes 4.08 s, the SURF algorithm takes 1.28 s, and the LP-SIFT algorithm takes 2.03 s. Since the SIFT algorithm takes more than 10^4^ s but does not return any matching result, the computation is terminated without returning a stitching time. In this computation set, the BRISK, SURF, and LP-SIFT algorithms require significantly less time than SIFT and ORB, which experiences a notable increase in the number of feature points as the image size is enlarged.

### 3.5. Images of Large Size with Translational Displacement

Large-sized images are frequently employed in mobile photography [65,66], satellite remote sensing [67,68], UAV aerial photography [23,69], and other fields. In our dataset, the image pairs in the campus view (translation) datasets have a large size. The size of the images is 3072 × 4096 pixels.

The stitching results are collectively shown in Figure 7a, incorporating those obtained using BRISK, SURF, and LP-SIFT. The corresponding parameters of the stitching process are summarized in Table 2. For campus view (translation) datasets, only three feature point detection algorithms, when combined with RANSAC, can successfully stitch the two images. Although the evaluation metrics computed from the SURF algorithm are the highest, the differences with those of BRISK and LP-SIFT are less than 1%. Therefore, in the stitching of campus view (translation) datasets, the three algorithms show a similar stitching effect: the BRISK algorithm takes 195.44 s, the SURF algorithm takes 6.52 s, and the LP-SIFT algorithm takes 4.49 s. Since the execution time for SIFT exceeded 10^4^ s without yielding any results, the computation was halted at that point. On the other hand, the ORB algorithm detected an excessive number of feature points, surpassing the computer’s memory capacity and leading to a stitching failure. Among the three algorithms, LP-SIFT exhibits orders of improvement in stitching efficiency. One key advantage of LP-SIFT is its adjustable interrogation window size, allowing for the control of the number of feature points even in extremely large images with a high signal-to-noise ratio (SNR). By applying a larger interrogation window size, LP-SIFT can effectively limit the number of feature points, resulting in a faster stitching process compared to other algorithms. This enhanced efficiency makes LP-SIFT a compelling choice for image stitching tasks, especially in scenarios where computational resources are limited.

### 3.6. Images of Large Size with Rotational Displacement

The SIFT algorithm has rotation invariance. In this section, we hope to demonstrate that the LP-SIFT algorithm also reserves the rotation invariance feature. The image pairs in the campus view (rotation) datasets with a large size of 3072 × 4096 pixels are applied. The image pairs are captured separately by the same camera, and are not artificially rotated from each other. SIFT, ORB, BRISK, SURF, and LP-SIFT algorithms were employed to detect feature points and feature description vectors. The RANSAC algorithm is further used to stitch the images. Figure 7b depicts the stitching results achieved by combining the SURF and LP-SIFT feature point detection algorithms with RANSAC, along with the stitching parameters summarized in Table 2. It is evident that both feature point detection algorithms can successfully stitch the two images. Relative to SURF, the LP-SIFT algorithm provides a better stitching effect according to the higher evaluation metrics, in addition to a shorter stitching time. The SURF algorithm takes 11.42 s, while the LP-SIFT algorithm takes 4.58 s, which is only 40% of the computation time of SURF. Similar to the results seen in Section 3.4, the SIFT algorithm requires an unacceptably long time for stitching, while the ORB and BRISK algorithms fail in stitching due to the overflow of memory. Therefore, LP-SIFT shows the capability of stitching images with rotational displacements, and is particularly fast for images with a large size.

### 3.7. Discussion

Overall, the LP-SIFT algorithm shows comparable evaluation metrics as the other stitching algorithms in all the five datasets. It has the same or even better robustness as other algorithms such as ORB and BRISK. Compared to the original SIFT method, the speed of feature point detection of LP-SIFT can be promoted by 109 times for small-size images and by orders of magnitude for larger images. The time consumption in SIFT is primarily attributed to the calculation of feature description vectors, particularly when dealing with many detected feature points, especially in the case of large images. ORB and BRISK face challenges due to the rapid increase in feature points with image size. This may not be a big deal if the computational resource is sufficiently large, e.g., a commercial workstation, but it could be an obstacle for applications in a portable computation system. The SURF algorithm is applicable for stitching images of different sizes, but the overall efficiency is relatively lower than that of LP-SIFT.

Relative to SIFT, ORB, BRISK, and SURF, LP-SIFT can flexibly adjust the interrogation window size to determine the multiscale local peaks. Therefore, the number of feature points can be well controlled without a significant increase with the image size, which is the advantage of replacing the Gaussian pyramid and difference pyramid feature point detection. This is why LP-SIFT can perform fast image stitching for large images. One may need to note that the algorithm of LP-SIFT is programed in MATLAB without acceleration by a GPU. If LP-SIFT is developed by C/C++ with GPU acceleration, the computation efficiency can be significantly promoted.

On the other hand, similar to the conventional image stitching techniques, the LP-SIFT algorithm faces challenge when dealing with periodic structures, or in the environments with weak texture features and a high noise (low signal-to-noise ratio) level. Furthermore, if the images have more small-scale contents (high-frequency components), more feature points on small scales may be inevitable. The size of the interrogation window may be reduced. Accordingly, the number of feature points and the computational time will also be increased.

## 4. Mosaic of Multiple Images without Prior Knowledge

In various application scenes, e.g., criminal investigation [70], remote sensing monitoring [68], and UAV aerial photography [69], images are probably fragmented with unknown positions, angles, and sequences that need to be restored. Thus, the mosaic strategy of stitching multiple images, which is more complex than the two-image case, is also crucial. Here we propose a mosaic strategy for combining multiple images without prior knowledge.

As illustrated in Figure 8, the first step involves using LP-SIFT to compute the homography matrix between each pair of images within a given dataset, which is stored in the matrix ($H_{M}$). This aims to find the transformation relationship between individual image pairs. The process can be parallelized using CPU computing to save computation time.

Subsequently, the number of nonzero elements in each row or column of $H_{M}$ is counted, and the reference image is selected based on the row or column with the highest count of nonzero elements. This process guarantees the reference image with the most neighbor images (the images can be stitched with the reference image) can be stitched first to reduce the subsequent iterations and save the time required to produce the mosaic. The images matched with the reference image are stitched in this round, leaving the unmatched images stored in the unmatched set. Then, in the second round, another reference image is selected from the unmatched set according to $H_{M}$. By repeating the process above, the mosaic is finally terminated until the number in the unmatched set reaches 0.

Figure 9a depicts the original image captured in this experiment. The image size is 6400 × 4270. Figure 9b shows the fragmented images randomly divided from the original image. The approach outlined in Figure 8 is employed to seamlessly merge the image fragments shown in Figure 9b. As shown in Figure 9c, the result is highly coincident with the original image in Figure 9a. The stitching time is 158.94 s, which is acceptable for a wide range of applications.

## 5. Conclusions

In this study, we propose a fast feature point detection algorithm, namely local-peak scale-invariant feature transform (LP-SIFT), which integrates the concept of local extremal points or image peaks at the multiscale level with SIFT. By integrating LP-SIFT and RANSAC in image stitching, significant improvements in stitching speed are achieved compared to the original SIFT method. Furthermore, LP-SIFT was evaluated against ORB, BRISK, and SURF for stitching images of varying sizes. Due to its adaptability in adjusting the interrogation window size, LP-SIFT demonstrates a minimal increase in the number of detected feature points with increasing image size, resulting in a noticeable reduction in stitching time, particularly for large-scale cases. Additionally, we also provide a strategy for seamlessly stitching multiple images using LP-SIFT without prior knowledge. It is anticipated that LP-SIFT will contribute to diverse application scenarios such as terrain mapping, biological analysis, and even criminal investigations.

## Figures and Tables

**Figure 1 sensors-24-05759-f001:**
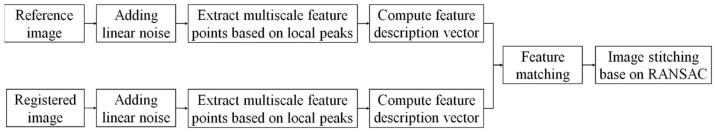
Framework of the LP-SIFT method.

**Figure 2 sensors-24-05759-f002:**
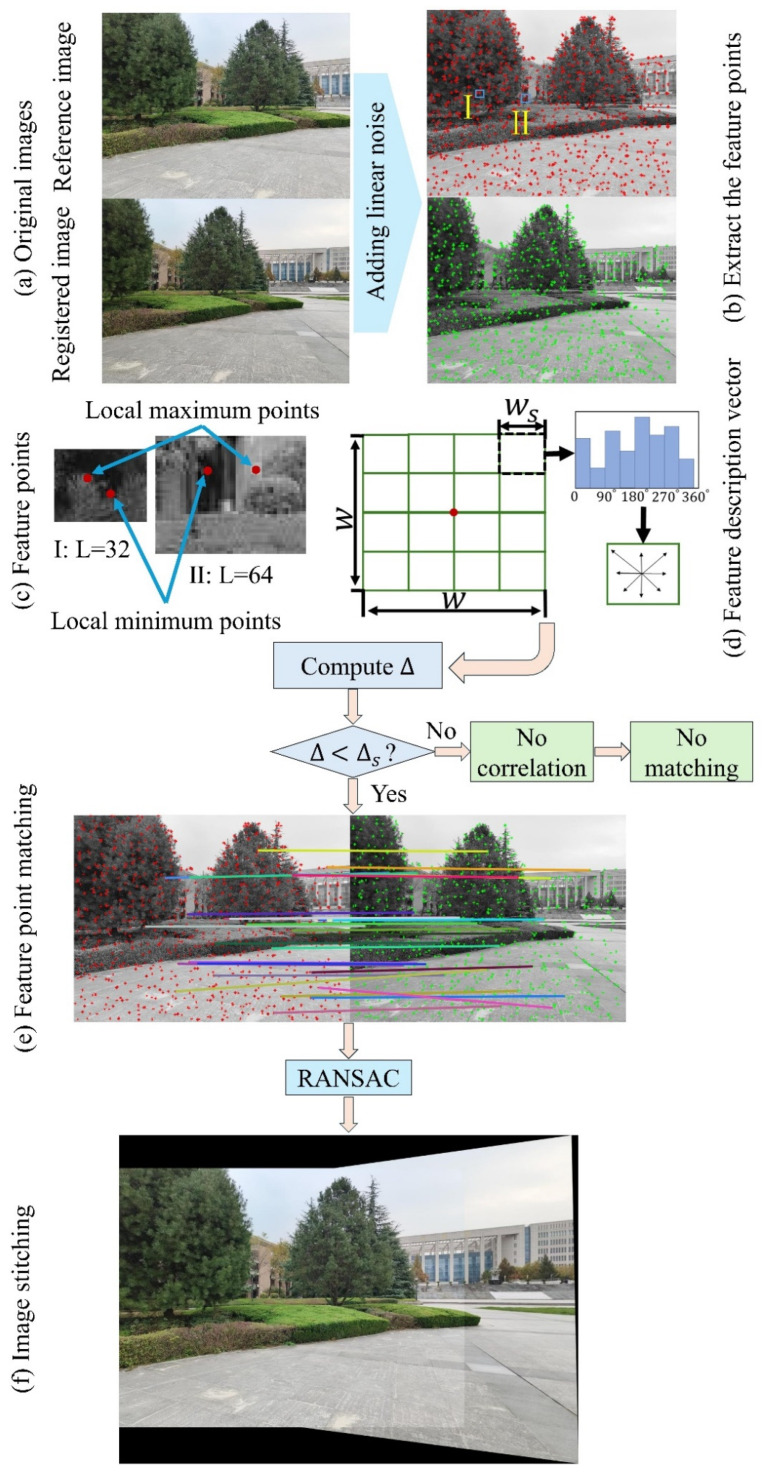
Diagram of the LP-SIFT method.

**Figure 3 sensors-24-05759-f003:**
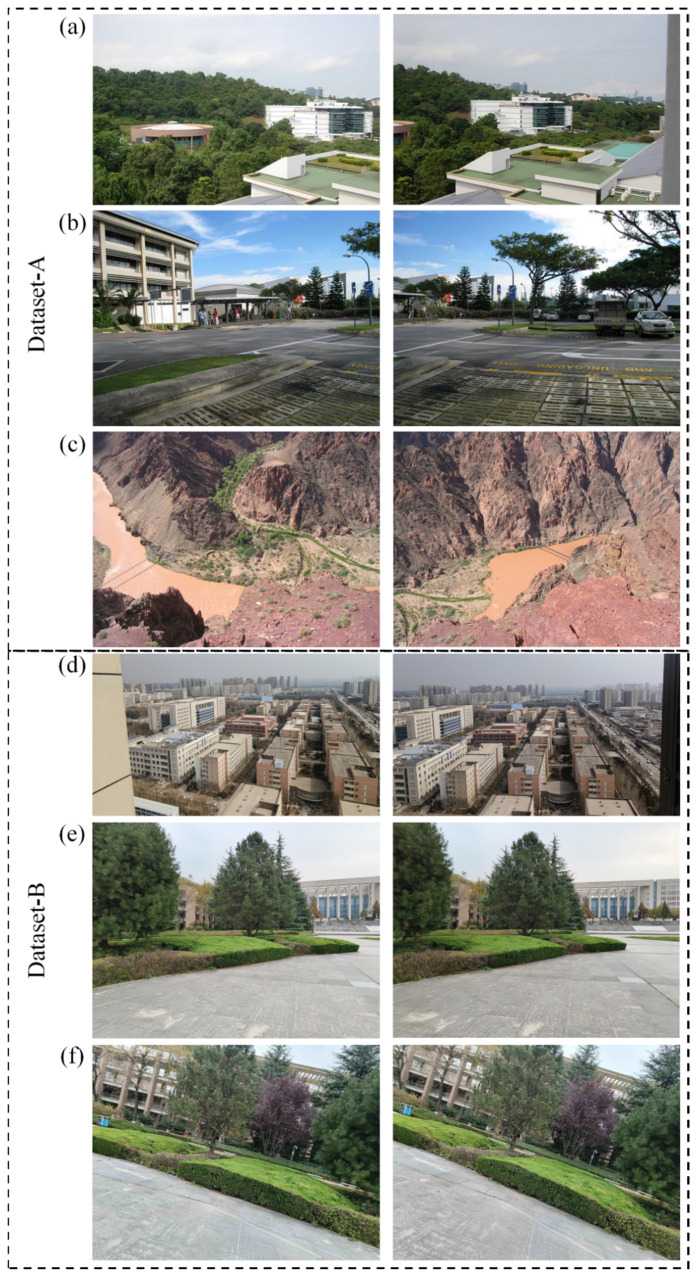
Datasets. Dataset-A: (**a**) mountain [52] dataset image pair, (**b**) street view [53] dataset image pair, (**c**) terrain [54] dataset image pair. Dataset-B: (**d**) building dataset image pair, (**e**) campus view dataset (translation) image pair, (**f**) campus view dataset (rotation) image pair.

**Figure 4 sensors-24-05759-f004:**
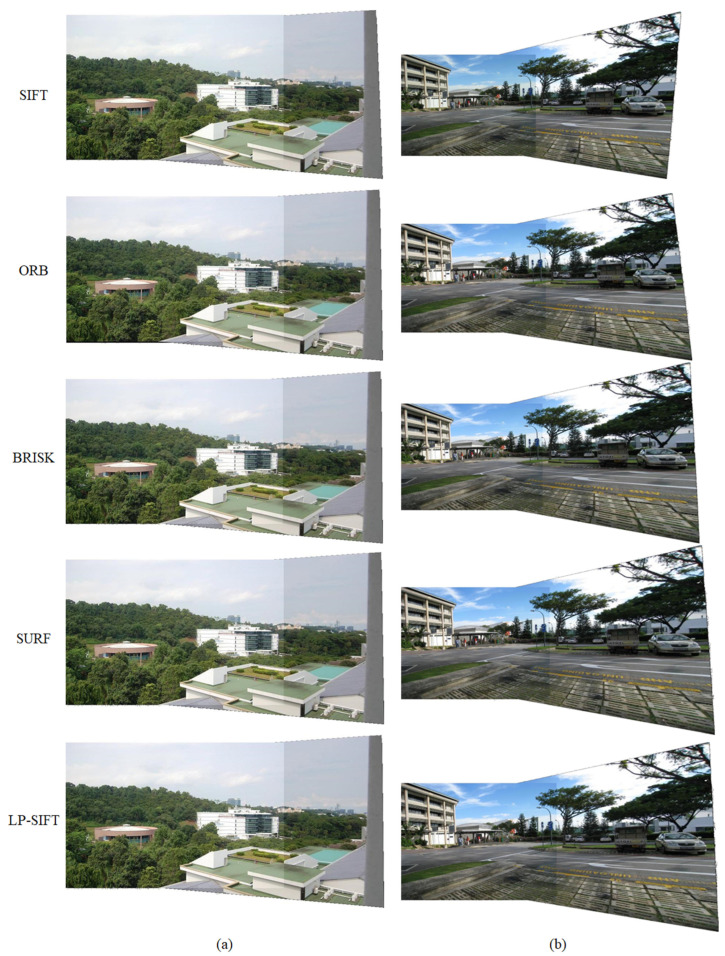
Stitching results of mountain dataset and street dataset. (**a**) Mountain dataset stitched by SIFT, ORB, BRISK, SURF, and LP-SIFT, respectively. In LP-SIFT, *L* = [32, 40]. (**b**) Street view dataset stitched by SIFT, ORB, BRISK, SURF, and LP-SIFT, respectively. In LP-SIFT, *L* = [32, 40].

**Figure 5 sensors-24-05759-f005:**
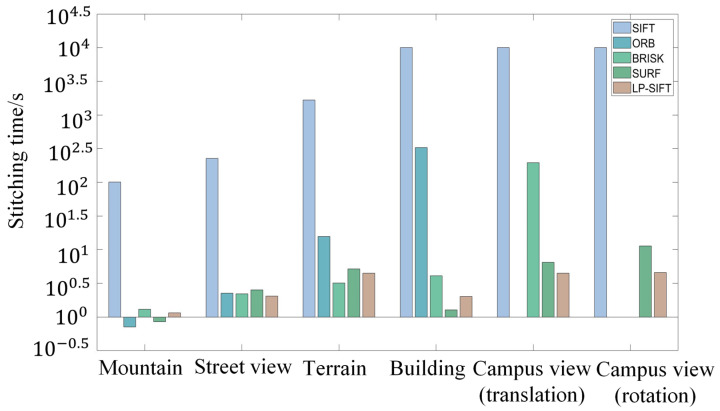
Comparison of the stitching times of 5 algorithms for different datasets.

**Figure 6 sensors-24-05759-f006:**
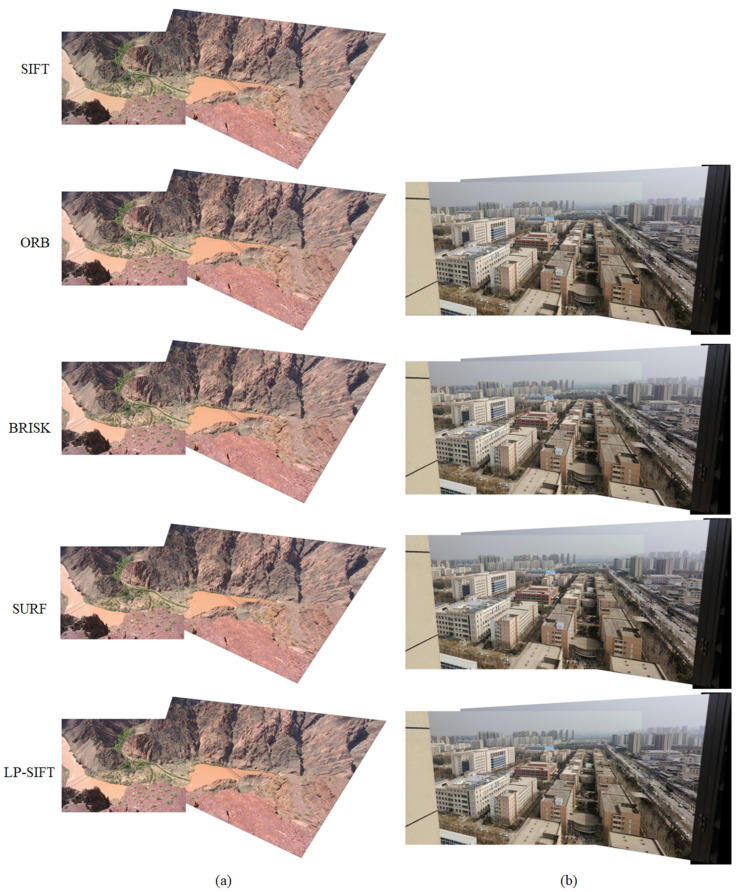
Stitching results of terrain dataset and building dataset. (**a**) Terrain dataset stitched by SIFT, ORB, BRISK, SURF, and LP-SIFT respectively. In LP-SIFT, *L* = [32,64]. (**b**) Building dataset stitched by ORB, BRISK, SURF, and LP-SIFT respectively. In LP-SIFT, *L* = [100,128].

**Figure 7 sensors-24-05759-f007:**
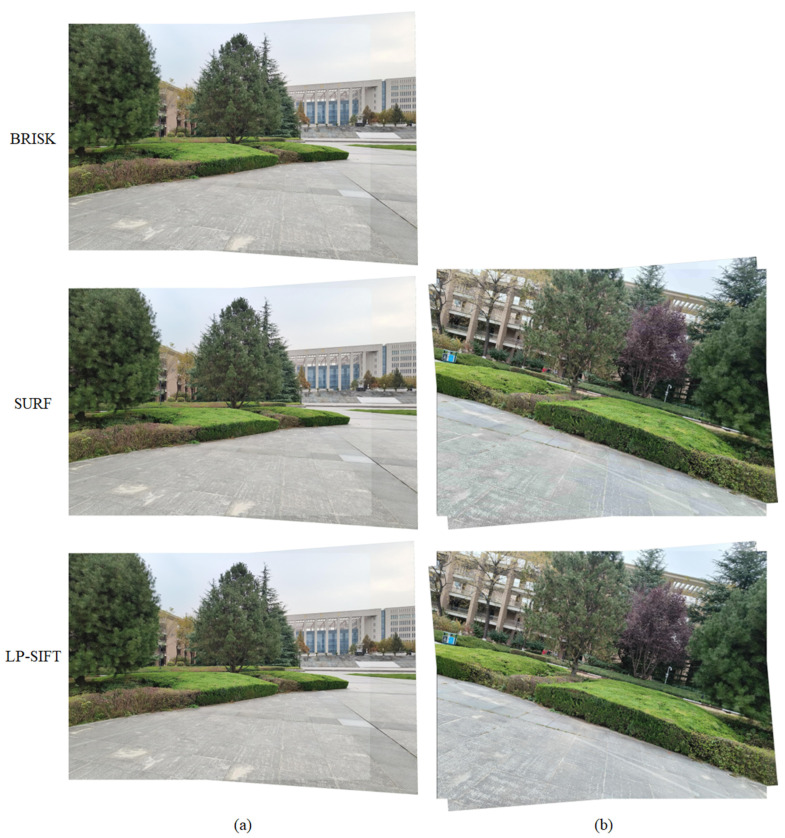
Stitching results of campus view dataset. (**a**) Campus view (translation) dataset stitched by BRISK, SURF, and LP-SIFT, respectively. In LP-SIFT, *L* = [256,512]. (**b**) Campus view (rotation) dataset stitched by SURF, and LP-SIFT, respectively. In LP-SIFT, *L* = [256,512].

**Figure 8 sensors-24-05759-f008:**
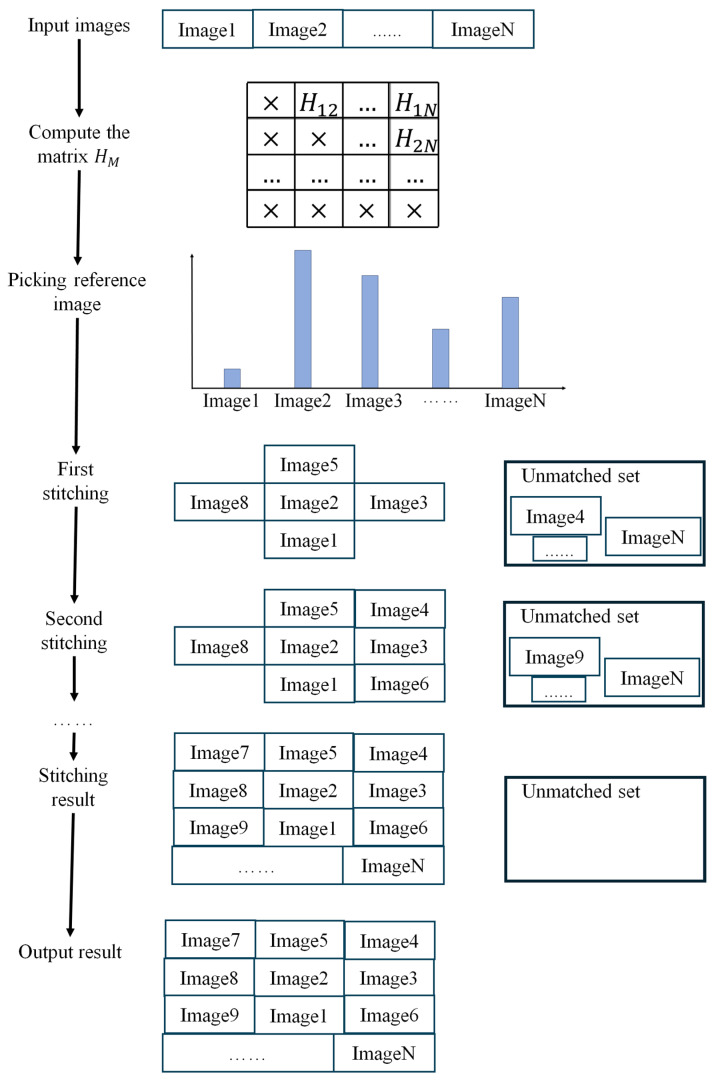
Schematic diagram of LP-SIFT image mosaic of multiple images without prior knowledge.

**Figure 9 sensors-24-05759-f009:**
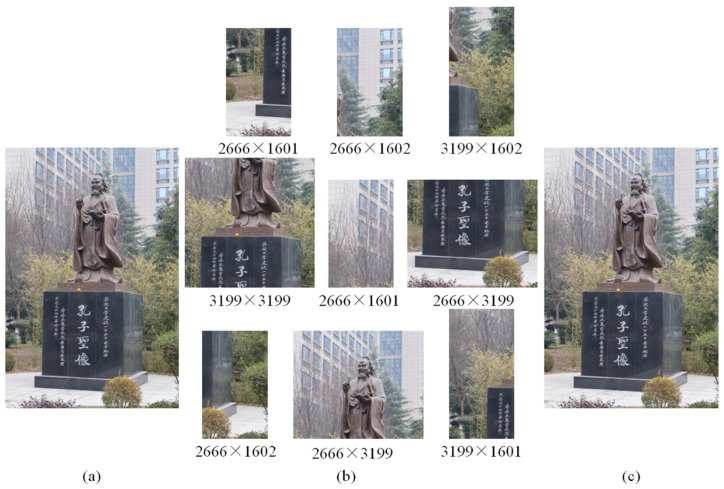
Mosaic of multiple images by LP-SIFT without prior knowledge, where *L* = [512,1024]. (**a**) Original image; the image size is 6400 × 4270. (**b**) The original image is stitched into different sizes and its position is shuffled, and its size is marked below the image. (**c**) Stitching result, and the stitching time is 158.94 s.

**Table 1 sensors-24-05759-t001:** Hardware and software specifications.

Hardware	Operation system	Windows 11 64-bit operating system
	Processor	Intel Core i9-12900
	Memory	64 GB
	Graphics card	NVIDIA GeForce RTX 3090
Software	Platform	MATLAB 2021a
	Library	Computer Vision Toolbox 10.0
Development environment	SIFT	MATLAB 2021a
	ORB	MATLAB 2021a
	BRISK	MATLAB 2021a
	SURF	MATLAB 2021a
	LP-SIFT	MATLAB 2021a

**Table 2 sensors-24-05759-t002:** Parameter setup of different feature point detection algorithms. To maintain the consistency for comparison, all of them were stitched by the RANSAC algorithm. The numbers marked by background colors show the minimal stitching time, the largest AG and SF of the algorithms respectively in the processing in each dataset.

Name	Image Size		Algorithm	Number of Feature Points		Number of Matched Pairs	Stitching Time (s)	AG	SF
Mountain	Small image	602 × 400	SIFT	1496	939	15	101.21	6.68	27.73
			ORB	10,505	6926	1132	0.71	6.60	27.34
			BRISK	1416	1122	75	1.30	6.51	27.20
			SURF	638	490	216	0.85	6.55	27.28
			LP-SIFT	487	493	31	1.16	6.54	27.28
Street view	Small image	653 × 490	SIFT	1948	2726	23	226.62	6.58	23.96
			ORB	11,363	15,597	541	2.27	7.17	25.85
			BRISK	2523	4104	59	2.22	6.70	24.39
			SURF	933	1149	137	2.54	6.63	24.10
			LP-SIFT	811	812	59	2.05	7.00	25.23
Terrain	Medium image	1024 × 768	SIFT	9495	10,368	134	1674.87	5.41	17.33
			ORB	3182	3182	2224	15.77	5.48	17.36
			BRISK	8149	8306	95	3.20	5.39	17.11
			SURF	2883	3037	204	5.16	5.49	17.75
			LP-SIFT	1847	1837	29	4.47	5.50	17.78
Building	Medium image	1080 × 1920	SIFT	×	×	×	>10^4^	×	×
			ORB	107,612	108,452	9720	327.25	6.57	24.29
			BRISK	14,660	15,428	605	4.08	6.56	24.33
			SURF	6123	5985	1780	1.28	6.58	24.35
			LP-SIFT	532	484	17	2.03	6.47	24.11
Campus view (translation)	Large image	3072 × 4096	SIFT	×	×	×	>10^4^	×	×
			ORB	1,025,750	927,050	Over size	×	×	×
			BRISK	104,981	94,657	3299	195.44	9.37	24.87
			SURF	27,790	25,465	6056	6.52	9.44	24.96
			LP-SIFT	418	403	29	4.49	9.37	24.88
Campus view (rotation)	Large image	3072 × 4096	SIFT	×	×	×	>10^4^	×	×
			ORB	1,326,389	1,332,929	Over size	×	×	×
			BRISK	158,247	164,035	Over size	×	×	×
			SURF	47,568	47,678	11,293	11.42	10.42	24.26
			LP-SIFT	422	429	22	4.58	11.74	27.37

## Data Availability

The data presented in this study are available on request from the corresponding author. The data are not publicly available due to the privacy. During the preparation of this work, the author(s) used ChatGPT 4.0 in order to check essay presentation, grammar, and spelling. After using this tool, the authors reviewed and edited the content as needed and take full responsibility for the content of the publication.

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
