# Peer review of "Local-Peak Scale-Invariant Feature Transform for Fast and Random Image Stitching"

_sensors, 2024, doi:10.3390/s24175759_

Round 1

Reviewer 1 Report

Comments and Suggestions for Authors

The paper proposes a new algorithm called Local-Peak Scale-Invariant Feature Transform (LP-SIFT) for faster image stitching, which improves upon the original SIFT method by using multiscale local peaks instead of the Gaussian pyramid. The LP-SIFT combined with RANSAC significantly increases the speed of stitching large images without prior knowledge, making it suitable for applications like terrain mapping, biological analysis, and criminal investigation.

Drawbacks of the Paper

  1. Computational Limitations:

    • The paper mentions that the LP-SIFT algorithm, though faster than traditional SIFT, is still programmed in MATLAB without GPU acceleration. This means that the current implementation may not be as efficient as it could be with further optimization and hardware acceleration​.
  2. Algorithm Robustness:

    • While LP-SIFT shows significant improvement in stitching speed and maintains accuracy, it is not as robust to variations in illumination and viewpoint changes as some other algorithms like ORB and BRISK. The paper acknowledges that these algorithms can handle these variations better, though they also face challenges with large images​​.
  3. Memory Consumption:

    • For very large images, ORB and BRISK algorithms can lead to memory overflow issues. While LP-SIFT mitigates some of these problems by controlling the number of feature points, it still requires a substantial amount of memory for very large datasets​.
  4. Lack of Deep Learning Integration:

    • The paper notes that recent trends are shifting towards deep learning-based approaches for image stitching. Although LP-SIFT optimizes traditional methods, it does not incorporate deep learning techniques, which might limit its applicability in cutting-edge research and applications​.
  5. Comparison Limitations:

    • The comparative analysis is done using MATLAB’s implementations of various algorithms, which might not be the most optimized versions available. The performance could vary significantly with different implementations or optimizations in other programming environments​.
  6. Limited Real-World Testing:

    • The datasets used for testing, while varied, might not cover all possible real-world scenarios. There might be specific conditions or edge cases in practical applications where LP-SIFT’s performance could differ from the reported results​.

Author Response

Authors’ reply to reviewer 1

We deeply appreciate the reviewers for reviewing the manuscript and giving us such valuable and helpful comments to improve our manuscript. All these comments have been carefully addressed point-by-point as follows. The reviewer’s original comments are in black and the authors’ replies are in blue. The red color represents the corresponding revision in the manuscript.

Comment:

  1. Computational Limitations:

The paper mentions that the LP-SIFT algorithm, though faster than traditional SIFT, is still programmed in MATLAB without GPU acceleration. This means that the current implementation may not be as efficient as it could be with further optimization and hardware acceleration.

Reply

1.We sincerely appreciate the comment. This study elucidates an analogy between image structures and the dissipation elements (DEs) in turbulence, subsequently applying the local peak concept from DEs to the Scale-Invariant Feature Transform (SIFT). We have developed a rapid feature point detection algorithm termed Local-Peak Scale-Invariant Feature Transform (LP-SIFT), grounded in multiscale local peaks and the scale-invariant feature transform methodology. By integrating LP-SIFT with RANSAC in image stitching, we have significantly enhanced the stitching speed compared to the original SIFT approach.

The aim of the this investigation is to demonstrate the efficacy and practicality of the method. We agree that in MATLAB without GPU acceleration, the performances of the methods have been significantly restricted. Our forthcoming efforts will focus on algorithm optimization and hardware acceleration in C#, with the aspiration of achieving greater efficiency.

Comment:

  1. Algorithm Robustness:

While LP-SIFT shows significant improvement in stitching speed and maintains accuracy, it is not as robust to variations in illumination and viewpoint changes as some other algorithms like ORB and BRISK. The paper acknowledges that these algorithms can handle these variations better, though they also face challenges with large images.

Reply

2.We sincerely appreciate the comment. This algorithm has been designed with considerations for both viewpoint and illumination changes during data set selection. As illustrated in Figure 3 (a - d), distinct differences in viewpoints are evident from the stitched results shown in Figures 4 and 5. Regarding illumination changes, the seams observed in the stitching results of each group, which arise from the varying illumination intensities of the images, underscore the algorithm’s robustness to such changes. From the various scenarios studied in this investigation, there are no indications that LP-SIFT exhibits weaker robustness to other algorithms such as ORB and BRISK. In addition, we introduce Average Gradient (AG) and Spatial Frequency (SF) as evaluation parameters. The parameters show that LP-SIFT algorithm is comparable and even better compared to the other algorithms such as ORB and BRISK. Therefore, LP-SIFT has the same robustness as other algorithms such as ORB and BRISK.

In the revised manuscript, we have added corresponding data to support the robustness of the algorithm in section 3.

Comment:

  1. Memory Consumption:

For very large images, ORB and BRISK algorithms can lead to memory overflow issues. While LP-SIFT mitigates some of these problems by controlling the number of feature points, it still requires a substantial amount of memory for very large datasets.

Reply

3.We sincerely appreciate the comment. To achieve a high stitching efficiency, we need to reduce the cost of time in every step, including the readout and transport time in a computer system. Therefore, a requirement of sufficiently large memory is inevitable, especially for exceptionally large datasets. The advantage of the algorithm proposed in this study is its capability to perform data stitching for larger datasets while operating within the constraints of limited computational resources, by memory management strategies and optimization of interrogation window size.

Comment:

  1. Lack of Deep Learning Integration:

The paper notes that recent trends are shifting towards deep learning-based approaches for image stitching. Although LP-SIFT optimizes traditional methods, it does not incorporate deep learning techniques, which might limit its applicability in cutting-edge research and applications.

Reply

4.We sincerely appreciate the comment. Despite the rapid advancement and predominant status of deep learning, traditional methods remain indispensable. At present, the internal mechanisms of most deep learning algorithms resemble a “black box,” rendering their specific functions challenging to elucidate. Furthermore, various deep learning models are typically tailored to particular scenarios, making them less versatile. Although the LP-SIFT algorithm presented in this study represents an enhancement of traditional methods, it allows for a clear analysis of its internal principles and offers novel insights into the development of deep learning models. Moreover, LP-SIFT method can be used to high-efficiently generate training sets for developing new deep learning methods, or hybrid methods which integrate traditional methods with deep learning models. Hence, it does not restrict its applicability in cutting-edge research and applications. We believe, the factor that determines what method is cutting-edge is performance only.

Comment:

  1. Comparison Limitations:

The comparative analysis is done using MATLAB’s implementations of various algorithms, which might not be the most optimized versions available. The performance could vary significantly with different implementations or optimizations in other programming environments.

Reply

5.We sincerely appreciate the comment. As in the reply to the first comment, the aim of this investigation is to demonstrate the efficacy and practicality of the LP-SIFT method. We agree that in MATLAB without GPU acceleration, the performances of the methods have been significantly restricted. Our forthcoming efforts will focus on algorithm optimization and hardware acceleration in C#, with the aspiration of achieving greater efficiency. We will further make comparisons with the other algorithms in C# again.

Comment:

  1. Limited Real-World Testing:

The datasets used for testing, while varied, might not cover all possible real-world scenarios. There might be specific conditions or edge cases in practical applications where LP-SIFT’s performance could differ from the reported results.

Reply

6.We sincerely appreciate the comment. For each algorithm, the performance will be different in different scenes. To the best of authors’ knowledge, there is no universal method that can cover all real-world scenarios to date. Algorithms always face problems when stitching periodic structures, weak texture feature structures and images with ultralow SNR. This is also applicable for LP-SIFT.

Reviewer 2 Report

Comments and Suggestions for Authors

This paper addresses the problem of fast feature extraction and matching for large-scale images by proposing a Local Peak Scale-Invariant Feature Transform algorithm (LP-SIFT). The LP-SIFT draws inspiration from the concept of turbulent dissipation elements in fluid mechanics, introducing a multi-scale local peak detection method to replace the Gaussian pyramid construction process in the original SIFT algorithm. The performance of LP-SIFT was compared across various image types and sizes. However, there are several concerns of the paper that need to be addressed:

1.      In the seventh line of the fifth paragraph of the Introduction, the paper described an analogy between image structures and the dissipation elements (DEs) in turbulence, subsequently applying the local peak concept from DEs to SIFT in the section II. However, there lacks an in-depth theoretical analysis of this analogy. The authors merely state this similarity without adequately explaining its rationality or providing strong support from mathematical or physical perspectives. And it is not mentioned again when describing the proposed method.

2.      The contribution of the paper needs to be strengthened.

3.      In section III, the comparison of different image stitching algorithms primarily relies on visual inspection, which is subjective and imprecise. As shown in Figures 3 and 4, the differences between various algorithms are difficult to discern. It is recommended that the authors introduce objective evaluation metrics such as Structural Similarity Index (SSIM) and Peak Signal-to-Noise Ratio (PSNR) to quantitatively compare the stitching differences between different methods.

4.      As evident from Table 2 and Figure 5, the LP-SIFT algorithm does not achieve the shortest processing time in some tasks, sometimes even underperforming compared to SURF or ORB algorithms. It needs in-depth analysis.

5.      From Table 2, it can be observed that the number of feature points detected by LP-SIFT does not change significantly as the image size increases, while other methods show notable differences. This might be the reason why LP-SIFT can achieve faster processing speeds when dealing with large-scale images. Would other methods also achieve better performance if they used RANSAC to limit the number of matching points?

6.      Why there are negative values for the mountain image data in Figure 5?

7.      What are the potential limitations of the LP-SIFT algorithm?

Author Response

Authors’ reply to reviewer 2

We deeply appreciate the reviewers for reviewing the manuscript and giving us such valuable and helpful comments to improve our manuscript. All these comments have been carefully addressed point-by-point as follows. The reviewer’s original comments are in black and the authors’ replies are in blue. The red color represents the corresponding revision in the manuscript.

Comment:

  1. In the seventh line of the fifth paragraph of the Introduction, the paper described an analogy between image structures and the dissipation elements (DEs) in turbulence, subsequently applying the local peak concept from DEs to SIFT in the section II. However, there lacks an in-depth theoretical analysis of this analogy. The authors merely state this similarity without adequately explaining its rationality or providing strong support from mathematical or physical perspectives. And it is not mentioned again when describing the proposed method.

Reply

1.We sincerely appreciate the comment. Turbulence is a complex, random phenomenon widely exists in fluid dynamics, quantum physics, geoscience and astrophysics etc. It consists of multiscale structures, exhibiting many similarities, including scaling similarity, spatial and temporal similarities (primarily on large scale). These features can be statistically described by the scaling properties of velocity structure function, i.e. the difference of two-point velocity on a spatial increment . Wang et al [1-3]found the scaling property of the structure of turbulence could be alternatively described by dissipative element (DE) analysis. The relationship of local extreme velocity points (max and min) in an interrogation window with a size of can be alternatively and approximately used to describe the local turbulent velocity field on scale . In other words, we can use the local extreme velocity points to approximately describe the feature of turbulent velocity structure and similarity on scale , instead of calculating the velocity structure function of all the velocity points.

An image can be considered as a field of light intensity that consists of many feature structures. Two images that can be stitched must have similar feature structures, i.e. spatial similarity between images or parts of images at least. Therefore, in this investigation, we attempt to use local extreme points of different scales to describe the feature of image. Although a strict mathematical analog between the turbulence flow field and image field has not been established, the stitching results indicate LP-SIFT is effective.

Comment:

  1. The contribution of the paper needs to be strengthened.

Reply

2.We sincerely appreciate the comment. The abstract of manuscript has been revised accordingly as

“Abstract:

Image stitching aims to construct a wide field of view with high spatial resolution, which cannot be achieved in a single exposure. Typically, conventional image stitching techniques, other than deep learning, require complex computation and thus computational pricy, especially for stitching large raw images. In this study, inspired by the multiscale feature of fluid turbulence, we developed a fast feature point detection algorithm named local-peak scale-invariant feature transform (LP-SIFT), based on the multiscale local peaks and scale-invariant feature transform method. By combining LP-SIFT and RANSAC in image stitching, the stitching speed can be improved by orders, compared with the original SIFT method. Benefit by the adjustable size of interrogation window, LP-SIFT algorithm demonstrates comparable or even less stitching time than the other commonly used algorithms, while reserving comparable or even better stitching results. Nine large images (over 2600×1600 pixels), arranged randomly without prior knowledge, can be stitched within 158.94 s. The algorithm is highly practical for applications requiring a wide field of view in diverse application scenes, e.g., terrain mapping, biological analysis, and even criminal investigation.”

Comment:

  1. In section III, the comparison of different image stitching algorithms primarily relies on visual inspection, which is subjective and imprecise. As shown in Figures 3 and 4, the differences between various algorithms are difficult to discern. It is recommended that the authors introduce objective evaluation metrics such as Structural Similarity Index (SSIM) and Peak Signal-to-Noise Ratio (PSNR) to quantitatively compare the stitching differences between different methods.

Reply

3.We sincerely appreciate the suggestion.

To evaluate the stitching results with SSIM and PSNR, we need to use a standard reference image as original image. After splitting it into pieces and stitching back, we can compare the original and stitched images to calculate SSIM[4] and PSNR[5].

However, in this investigation, all the image pairs are captured separately. It is unattainable for us to evaluate SSIM and PSNR after stitching. Instead, Average Gradient (AG)[6] and Spatial Frequency (SF)[7] can be used for evaluate image stitching without a reference image. Therefore, we adopted these three methods to evaluate the stitching results.

For better clarity, the subsection 3.2 on page 8 of manuscript have revised as follows

“3.2. Evaluation metrics

Since the data sets selected in this study are all captured from actual scenes without a reference image, it is difficult to use structural similarity (SSIM) [56] and peak signal-to-noise ratio (PSNR) [57] to evaluate the image stitching results. To compare the stitching results by different methods, the indicators such as average gradient (AG) [58] and spatial frequency (SF) [59] are employed.

AG reflects the detail contrast and texture transformation in the image and can be used to measure the quality of the fused image. In image stitching, the larger the AG is, the better the stitching quality is. AG can be defined as follows [58]:

where  represents the number of rows of the image,  represents the number of columns of the image.

SF is another evaluation metric reflecting the change rate of image gray level. The larger the SF is, the clearer the image is, particularly for the image fusion after stitching. SF is calculated as follows [59]:

Comment:

  1. As evident from Table 2 and Figure 5, the LP-SIFT algorithm does not achieve the shortest processing time in some tasks, sometimes even underperforming compared to SURF or ORB algorithms. It needs in-depth analysis.

Reply

4.We sincerely appreciate the comment. For some small and medium-sized images stitching, LP-SIFT algorithm stitching time is not the least, the reason is that LP-SIFT algorithm is an improvement of SIFT algorithm, retains the feature description method of SIFT algorithm, only improves the feature point extraction process.

We next analyze the descriptors of SIFT, ORB and SURF in detail. For the SIFT algorithm, the descriptor is to calculate the histogram of gradient orientation in the area around the key point according to the scale and direction of the key point. To traverse each point around the key point, the calculation is very costly. It uses BRIEF descriptor generation to encode the image by learning a precomputed binary pattern to generate the required descriptor. For SURF algorithm, it uses the Haar wavelet response of the local image to construct the feature descriptor. The proposal of ORB and SURF is to accelerate the SIFT algorithm, and LP-SIFT is also an improvement of SIFT, but the advantages of SIFT algorithm are reflected in the image stitching of large size, so for small and medium-sized image stitching, the time is not the shortest and normal. At the same time, thank you again for your comments. Your comments are also new inspiration for us. We will consider changing the calculation form of the descriptor to further improve the algorithm in the future.

Comment:

  1. From Table 2, it can be observed that the number of feature points detected by LP-SIFT does not change significantly as the image size increases, while other methods show notable differences. This might be the reason why LP-SIFT can achieve faster processing speeds when dealing with large-scale images. Would other methods also achieve better performance if they used RANSAC to limit the number of matching points?

Reply

5.We sincerely appreciate the comment. Firstly, RANSAC algorithm is designed to fit the correspondence between feature points rather than to restrict the number of matching points, thereby it does not impose a limit on the number of feature points. In this investigation, for comparison, all the RANSAC algorithms for different stitching methods have the same setting. LP-SIFT algorithm shows a faster processing speed is due to the employment of adjustable window.

Comment:

  1. Why there are negative values for the mountain image data in Figure 5?

Reply

6.We sincerely appreciate the comment. The Y-axis in Figure 5 was plotted in a log coordinate which may cause misleading. For better clarity, we have revised the Y-axis in Figure 5 as follows

Figure 5. Comparison of the stitching times of 5 algorithms for different datasets.

Comment:

  1. What are the potential limitations of the LP-SIFT algorithm?

Reply

7.We sincerely appreciate the comment. For LP-SIFT algorithm, its performance will become poor when dealing with periodic structures, weak texture feature structures and structures with high noise (low signal-to-noise ratio). Besides, if the images have more small scale contents (high frequency components), more feature points on small scales may be inevitable. The window size may be reduced. Accordingly, the number of feature points will be increased and the computational time will be increased as well.

For better clarity, we have revised the third paragraph of “3.7. Discussion” on page 13 as follows

“On the other hand, similar as the conventional image stitching techniques, LP-SIFT algorithm faces challenge when dealing with periodic structures, or in the environments with weak texture features and high-noise (low signal-to-noise ratio) level. Besides, if the images have more small-scale contents (high frequency components), more feature points on small scales may be inevitable. The size of interrogation window may be reduced. Accordingly, the number of feature points and the computational time will be increased as well.”

Once again, we sincerely appreciate the reviewer’s instructive comments. The manuscript has been carefully revised according to the comments. Hope the current version has addressed the reviewer’s concern and can be acceptable for publication.

[1]N. Peters and L. Wang. Dissipation Element Analysis of Scalar Fields in Turbulence, Comptes Rendus. Mécanique, 2006: 493-506. https://doi.org/10.1016/j.crme.2006.07.006.

[2]L. Wang and N. Peters. Length-Scale Distribution Functions and Conditional Means for Various Fields in Turbulence, Journal of Fluid Mechanics, 2008: 113-138. https://doi.org/10.1017/s0022112008002139.

[3]L. Wang and N. Peters. The Length-Scale Distribution Function of the Distance between Extremal Points in Passive Scalar Turbulence, Journal of Fluid Mechanics, 2006. https://doi.org/10.1017/s0022112006009128.

[4]Z. Wang, A. C. Bovik, H. R. Sheikh and E. P. J. I. t. o. i. p. Simoncelli. Image Quality Assessment: From Error Visibility to Structural Similarity, 2004: 600-612.

[5]A. Hore and D. Ziou. Image Quality Metrics: Psnr Vs. Ssim, 2010 20th International Conference on Pattern Recognition, 2010: 2366-2369. https://doi.org/10.1109/icpr.2010.579.

[6]J. Wu, H. Huang, Y. Qiu, H. Wu, J. Tian and J. Liu. Remote Sensing Image Fusion Based on Average Gradient of Wavelet Transform, IEEE International Conference Mechatronics and Automation, 2005, 2005: 1817-1821.

[7]R. Heilbronner and S. Barrett. Image Analysis in Earth Sciences: Microstructures and Textures of Earth Materials. 129 Springer Science & Business Media, 2013.

[8]C. E. J. T. B. s. t. j. Shannon. A Mathematical Theory of Communication, 1948: 379-423.

Round 2

Reviewer 1 Report

Comments and Suggestions for Authors

I think that the actual presentation of this article is acceptable 

Reviewer 2 Report

Comments and Suggestions for Authors

The manuscript has been revised according to the suggestions and further strengthened the research contributions. 

I have no more suggestions.